# Investigations on Anticancer Potentials by DNA Binding and Cytotoxicity Studies for Newly Synthesized and Characterized Imidazolidine and Thiazolidine-Based Isatin Derivatives

**DOI:** 10.3390/molecules27020354

**Published:** 2022-01-06

**Authors:** Nasima Arshad, Muhammad Ismail Mir, Fouzia Perveen, Aneela Javed, Memona Javaid, Aamer Saeed, Pervaiz Ali Channar, Shahid Iqbal Farooqi, Saad Alkahtani, Jamshed Anwar

**Affiliations:** 1Department of Chemistry, Faculty of Sciences, Allama Iqbal Open University, Islamabad 44000, Pakistan; overlord.scorpion6@gmail.com (M.I.M.); shahidfarooqi2015@gmail.com (S.I.F.); 2Research Center for Modeling and Simulations, National University of Sciences and Technology (NUST), Islamabad 44000, Pakistan; fouzia_qau81@yahoo.com; 3Healthcare Biotechnology Atta-ur-Rehman School of Applied Biosciences, National University of Sciences and Technology (NUST), Islamabad 44000, Pakistan; javedaneela19@gmail.com; 4Department of Chemistry, Quaid-i-Azam University, Islamabad 45320, Pakistan; memona.javaid06@gmail.com (M.J.); mrpervaiz@gmail.com (P.A.C.); 5Department of Zoology, College of Science, King Saud University, Riyadh 12546, Saudi Arabia; salkahtani@ksu.edu.sa; 6Department of Chemistry, University of Lancaster, Lancaster LA1 4YB, UK; j.anwar@lancaster.ac.uk

**Keywords:** isatin derivatives, DNA binding studies, binding correlations, cell line activity, anticancer drug candidacy

## Abstract

Imidazolidine and thiazolidine-based isatin derivatives (IST-01–04) were synthesized, characterized, and tested for their interactions with ds-DNA. Theoretical and experimental findings showed good compatibility and indicated compound–DNA binding by mixed mode of interactions. The evaluated binding parameters, i.e., binding constant (K_b_), free energy change (ΔG), and binding site sizes (n), inferred comparatively greater and more spontaneous binding interactions of IST-02 and then IST-04 with the DNA, among all compounds tested under physiological pH and temperature (7.4, 37 °C). The cytotoxic activity of all compounds was assessed against HeLa (cervical carcinoma), MCF-7 (breast carcinoma), and HuH-7 (liver carcinoma), as well as normal HEK-293 (human embryonic kidney) cell lines. Among all compounds, IST-02 and 04 were found to be cytotoxic against HuH-7 cell lines with percentage cell toxicity of 75% and 66%, respectively, at 500 ng/µL dosage. Moreover, HEK-293 cells exhibit tolerance to the increasing drug concentration, suggesting these two compounds are less cytotoxic against normal cell lines compared to cancer cell lines. Hence, both DNA binding and cytotoxicity studies proved imidazolidine (IST-02) and thiazolidine (IST-04)-based isatin derivatives as potent anticancer drug candidates among which imidazolidine (IST-02) is comparatively the more promising.

## 1. Introduction

Cancer, the second leading cause of death globally, is basically related to the abnormal functioning of deoxyribonucleic acid (DNA). Drugs are considered to destroy the cancerous cells or reduce their size [1]. DNA of cancerous cells also replicates by making more copies of itself. The most effective anticancer drugs should have the ability to stop this replication by damaging the DNA of cancer cells, but the drugs often also affect healthy cells, resulting in severe side effects. Hence a non-reversible type of binding of a drug molecule with DNA is unsafe, while non-covalent reversible type binding is relatively safer with the least cytotoxic effects on healthy cells. Therefore, there is still a need to identify chemical molecules that could bind with DNA reversibly and which could help to inhibit cancer with minimal or no cytotoxic effects on the cells of healthy parts of the body. In vitro compound–DNA binding studies via non-covalent interactions, including electrostatic interactions, groove binding, and intercalation [2], are very informative and helpful for the investigation of anticancer potential of a drug-like molecule in terms of binding modes and binding parameters [3,4,5].

Among various organic classifications, heterocyclic compounds play an important role in medicinal chemistry and in rational drug design [6]. Isatin (1H-indole-2,3-dione) (Figure 1) is a heterocyclic compound that contains a bicyclic ring, a benzene ring fused along pyrrolidine-2,3-dione ring. Isatins, a member of the indoline class of organic compounds that are present naturally in California prunes [7], belong to a leading group of heterocycles which are equipped with extraordinary biological activities such as anti-tuberculosis activity [8], anti-HIV activity [9], anti-convulsant activity [10], anti-microbial activity [11], antiglycation activity [12], anti-inflammatory activity [13], anti-malarial activity [14], and mainly anticancer activities [15,16,17].

Various natural and synthetic drug molecules consisting of isatin nucleus are clinically recognized agents. For example, 5-fluoro-3-substitued isatin named as sunitinib (Figure 2A) (sutent^®^) is FDA approved for GIST (gastrointestinal stromal tumors) and RCC (renal cell carcinoma) [18]. A 5-substituted hydroxamic acid derivative of isatin (Figure 2B) has been reported for histone deacetylases (HDAC) inhibitors which is also effective against cervical tumor cells [19]. Hybrids of isatin–pyrazole benzenesulphonamide (Figure 2C) were analyzed as inhibitors of carbonic anhydrase enzyme, which is a metalloenzyme involved in the pH buffering of intra and extracellular spaces [20]. Quinoline hybrids of isatin (Figure 2D) have been reported to exhibit significant anti-tuberculosis activity [21]. Isatin, when attached with thiazoles (Figure 2E), exhibited remarkable anti-HIV activity and were disclosed as HIV reverse transcriptase dual inhibitors [22]. Hydrazonoindolin-2-one derivatives are considered safer anticancer drugs for their anti-proliferative activity against human adenocarcinoma (A-549), human colorectal adenocarcinoma (HT-29), and human breast carcinoma (ZR-75) cell lines [23,24]. Besides antiviral [25,26] and tuberculostatic [27] activities, various isatin derivatives have been reported for anticancer and cytotoxic activities [28,29,30,31,32].

Multipurpose pharmacological activities of isatin moiety to be used as precursor for drug synthesis is the main motivation to develop four novel imidazolidine and thiazolidine-based isatin derivatives, namely (Z)-1-benzyl-5-chloro-3-(2-(4-(2-oxo-2H-chromen-3-yl)thiazol-2-yl)hydrazono) indolin-2-one (IST-01); (Z)-ethyl 2-(3-(1-benzyl-5-chloro-2-oxoindolin-3-ylideneamino)-4-oxo-2-thioxoimidazolidin-1-yl)acetate (IST-02); (Z)-1-benzyl-5-chloro-3-(2-(4-(3-oxo-3H-benzo[f]chromen-2-yl)thiazol-2-yl)hydrazono) indolin-2-one(IST-03); and (Z)-ethyl 2-(2-(1-benzyl-5-chloro-2-oxoindolin-3-ylidene)hydrazinyl)thiazole-4-carboxylate(IST-04) which contain three substituted 1-benzyl-5-chloro-indolin-2-one nuclei. Here, we report on their synthesis, characterizations, and anticancer potential as investigated by DNA binding and cytotoxicity studies.

## 2. Results and Discussion

### 2.1. Chemistry of Synthesized Compounds

Four new hydrazonoindolin-2-one compounds (IST-01–04) were synthesized with a general formula of [3-(hydrazono)1-benzyl-5-chloro-indolin-2-one]. Three different strategies were adopted to produce such derivatives (Figure 3). In the first strategy (Figure 3A), we focused on big lipophilic groups (coumarins and benzocoumarins) in the thiazole ring in the lead structure to achieve the target compounds IST-01 and IST-03. In the second strategy (Figure 3B), a hydrophilic moiety was utilized to replace the lipophilic groups to obtain the target structure IST-04. In the third strategy (Figure 3C), a bio-isosteric tactic was utilized to replace the thiazole ring by 2-thioxo-1,3-imidazolidine-4-one to obtain the target compound IST-02.

The final products were obtained by performing the reactions as shown in Figure 1. The reagent 5-chloroisatin was treated with benzyl bromide to obtain product (3), which was further reacted with thiosemicarbazide to obtain the Schiff base product. In the last step, the Schiff base product (5) was reacted with alpha-halogenated compounds to yield desired compounds 1-benzyl-5-chloro-3-(2-heteroarylhydrazono) indoline-2-ones (IST-01–04).

FT-IR analysis revealed that all synthesized compounds expressed characteristic absorption band for N-H at 3233 cm^−1^, C-H aromatic at 3119 cm^−1^, C=O of carbonyl at 1738 cm^−1^, and C=N of imine at 1549 cm^−1^. The ^1^H-NMR spectra of IST-01 showed NH proton at 13.34 ppm, coumarin proton at 8.63 ppm, thiazoline proton at 8.10 ppm. For IST-02, the ^1^H-NMR spectra showed isatin proton at 8.03 ppm, phenyl proton as a multiplet at 7.34–7.28 ppm, thiohydantoin proton at 4.97 ppm. For IST-03, ^1^H-NMR spectra showed N-H proton at 12.88 ppm, thiazolidine proton as a multiplet at 7.58–7.25 ppm. For IST-04,^1^H-NMR spectra showed N-H proton at 13.24 ppm, thiazoline proton at 7.79 ppm. ^13^C-NMR (75 MHz, CDCl_3_) in δ (ppm), for IST-01 showed 165.61 (thiazole, C=N), 161.63 (coumarin C=O), 152.99 (isatin C=O), 145.08 (imine, C). For IST-02, spectra showed 172.00 (thiohydantoin, C=S), 171.36 (thiohydantoin, C=O), 166.21 (ester C=O), 163.97 (isatin, C=O), 148.08 (imine, C). For IST-03, spectra showed 180.00 (thiazole, C=N), 161.63 (coumarin C=O), 160.80 (isatin C=O), 141.33 (imine, C). For IST-04, spectra showed 166.80 (ester, C=O), 161.35 (thiazole, C=N), 161.27 (isatin C=O), 144.21 (imine, C). All FT-IR, ^1^H-NMR, and ^13^C-NMR data was in good arrangement. The respective spectra are provided in Appendix A.

### 2.2. Theoretical Investigations for Compound–DNA Binding

#### 2.2.1. DFT Studies

The density functional theory (DFT) is an important theoretical method that could help in a better understanding of chemical and optical properties of compounds and their complexes. The electron density distributions and energy levels of isatin complexes with HOMO and LUMO are given in Figure 4. Theoretically, the orbital energy difference between HOMO and LUMO is named the HOMO–LUMO gap (DH–L). The optical band gap (E_g_) is the difference between the ground (S_0_) and excited state (S_1_) rather than the difference between orbital energies of HOMO and LUMO. Experimentally, the optical band gap (E_g_) obtained from the spectra is excitation or the lowest transition energy from the ground state to the excited state (first dipole state), considering only an electron, which is stimulated from the HOMO to the LUMO. The HOMO–LUMO energy gap decreases in the order as: IST-01 = IST-03 > IST-04 > IST-02.

The optimization energy and DH-L energy for IST-02 is the least, which demonstrates that it is the most reactive complex amongst other. The energy values for the S_0_ and S_1_ are shown in Table 1. The attached electron-donating groups, thiazole and 4-oxo-2-thioxoimidazolidin, withdraw the electron density toward the middle of the substituents. Similarly, for the complex IST-01 and IST-04, electron-donating group thiazole pulls the electron density towards the substituent. From the DFT studies, it can be concluded that the IST-02 conduction of electrons from the ground state to the excited state is comparatively easier due to the least energy gap. Moreover, the increase in electron-withdrawing capability results in lowering of the HOMO–LUMO gap.

#### 2.2.2. Molecular Docking Studies

To grasp the idea of modes of DNA binding and to identify the sites where a compound could bind in major or minor grooves of DNA, molecular docking is implemented as a valuable technique in theoretical drug design [33,34]. Molecular docking was performed by subjecting DNA to isatin compounds, with a number of docking poses per compound where each pose was assigned a score by MOE. The conformational search has been performed to identify the most favorable binding conformation. Best poses having most favorable interactions for all the derivatives were selected and shown in Figure 5. IST-01 furnished no hydrogen bonding interaction but hydrophobic interactions with the core of DNA, whereas the other three compounds developed H-bonding with the DNA base pairs. IST-02 developed a single H-bond with its receptor. However, IST-03 and IST-04 exhibited 2 H-bonds with the DNA base pairs. From the interaction pattern and binding constants (*K_b_*) calculated through molecular docking, it can be established that hydrogen bond interactions along with hydrophobic interaction are responsible for the highest *K_b_* value of the IST-02. When the MOE predicted conformation of IST-02, IST-03, and IST-04 was matched with the conformations obtained from an X-ray diffractometer, the hydrogen bonding mode with DNA was found to be the same.

IST-01 did not show any hydrogen bonding with the DNA helix. However, the most important DNA base pairs, i.e., DT7, DT8, DC9, DG10, DA17, DA18, DT19, DT20, and DC21, could interact with IST-01 and might be responsible for its better biological role. IST-02 depicted hydrogen bonding between its =O and –H atom of the base pairs guanine DG (B16). Other important residues that interacted with IST-02 included DC(A9), DG(B10), DC(A11), DG(B12), DA(B17), DA(B18), and DT(B19). A portion of IST-02 intercalated between DA(B17) and DA(B18) base pairs of the DNA. IST-03 depicted 2 hydrogen bonds: one with DA(B17) and the second with DG(B16). Other important residues responsible for binding of IST-03 within the groove included: DC(A9), DG(B10), DC(A11), DG(A12), DA (B18), and DT(B19). IST-04 displayed two hydrogen bonds associated with –H atom of IST-04 with –O of adenine DA (B17) and –N atom of IST-04 with –H atom of guanine DG(B16). Other important interacting sites included DG(B14), DC(A11), DC(B15), and DA(B18). A visualization of interactions showed partial intercalation and groove-binding modes for all isatin compounds. Based on scoring functions obtained for best docking pose, the binding constant was calculated. The binding strength order in term of binding constant was evaluated as: IST-02 > IST-04 > IST-03 > IST-01. The values are shown in Table 2. The highest *K_b_* values for IST-02 demonstrated its greater interaction with DNA base pairs.

### 2.3. Experimental Investigations for Compound–DNA Binding

#### 2.3.1. DNA Binding Studies by UV-Visible Spectroscopy

UV-visible spectrophotometry is a prime procedure to investigate compound–DNA binding. The variations in the UV-visible spectrum of a compound upon DNA addition could help identify the interactions involved between a compound and DNA.

The four isatin derivatives (IST-01–IST-04) have been evaluated spectrophotometrically to obtain an understanding of their interactions with ds-DNA. The spectra of all the derivatives were recorded, separately, at different concentrations and shown in Appendix A, while graphs were drawn for the absorbance at various concentration and provided in Appendix A. The molar extinction coefficients (ε; M^−1^cm^−1^) were evaluated from the slopes and the values obtained for IST-01–IST-04 were found to be 38,933, 20,750, 42,767, and 14,533, respectively. These values predicted π-π* transitions in all compounds in the λ_max_ range of 380–590 nm.

For DNA binding interactional studies, DNA titrations were performed, separately, into the fixed concentrations of each compound under physiological pH (7.4) and temperature (37 °C). The spectral responses before and after DNA titrations are shown in Figure 6. The peak intensity was observed as the drop in absorbance and the trend was found to be similar for all compounds after DNA titration. The percent drop in the absorption peak intensity was calculated by the following equation.
(1)$$H\%=\frac{A_{free}-A_{bound}}{A_{free}}\times 100$$

The resultant hypochromic effect was found to be 84.94% for IST-01, 65.88% for IST-02, 83.22% for IST-03, and 75.50% for IST-04 followed by a blue shift of magnitude 4.5 nm, 1 nm, and 5.5 nm for IST-01, IST-03, and IST-04, respectively, and a red shift of magnitude 10 nm for IST-02. Generally, hypochromism and a significant blue/red shift are the parameters that arise due to the change in the structure of DNA, with it reported for the intercalation mode of compound–DNA binding [3,35,36,37,38,39]. The spectral observation and related data showed that all four compounds interacted with DNA via intercalative binding [3,36,37].

Generally, the Benesi–Hildebrand and van’t Hoff equations were used to evaluate the binding constant and Gibbs free energy change, respectively.
(2)$$\frac{A_{o}}{A-A_{o}}\;=\;\frac{\varepsilon_{G}}{\varepsilon_{H-G}-\varepsilon_{G}}+\frac{\varepsilon_{G}}{\varepsilon_{H-G}-\varepsilon_{G}}\frac{1}{K_{b}\left[DNA\right]}$$
ΔG = −RT ln*K_b_*(3)

Here, *A_o_* and *A* are the absorbance of the compound in the absence and presence of DNA, respectively, and *ε_G_* and *ε_H-G_* are the molar extinction coefficients of free compound and the compound–DNA complex, respectively. The binding constant (*K_b_*) was calculated from the intercept-to-slope ratio from the plot of *A_o_/A* − *A_o_* vs. 1/[*DNA*] (Figure 7). The binding constant values were further employed in van’t Hoff equations to obtain the values for Gibbs free energy change (ΔG). Both binding parameters (*K_b_* and ΔG) are provided in Table 3. The order of magnitude for *K_b_* was evaluated to be 10^3^ which reflects spontaneous and substantial binding of the compounds with DNA, with the same order of binding constant reported for some intercalating molecules [3,38,39].

#### 2.3.2. DNA Binding Studies by Fluorescence Spectroscopy

Similar DNA titrations were carried out under identical conditions (pH 7.4, 37 °C) as those set for UV-visible experiments. All compounds were found to be luminescent; hence, their fluorescence spectra were recorded separately and then DNA titrations were carried out by adding different DNA concentrations directly into the fixed concentration of each compound. The fluorescence responses are shown in Figure 8. A remarkable rise in peak intensity was observed in the compounds’ spectra when DNA was added up to 90 µM. This enhancement in the emission intensity was found to be 8.88, 4.18, 3.67, and 2.54 times higher for IST-01–04, respectively, after the addition of maximum DNA concentration and revealed interaction between the tested compounds and DNA. The literature has reported such spectral changes in compound–DNA spectra as interaction via intercalative binding [3,40].

Fluorescent enhancement also indicated that the molecule is strictly shackled at the binding site and there is no hydrophobic interaction between the nitrogenous bases of the DNA and water molecules. So, after excitation, the masking of the fluorophore and the interaction between stacked nitrogenous bases of the DNA decreases the vibrational modes that contribute towards the increase in fluorescence peak intensity [40].

The fluorescence intensity data of each compound before and after DNA titration was used in the following equation to evaluate the binding constant and binding site size [41].
(4)$$\log\frac{F-F_{o}}{F_{}}\;=\log K_{b}\;+n\log\left[DNA\right]$$
where ‘*K_b_*’ and ‘*n*’ are the binding constant and binding site size, respectively. From the plots of log [*F* − *F_o_/F_o_*] vs. log [*DNA*], (Figure 9), ‘*K_b_* ’and ‘*n*’ of each compound–DNA complex were evaluated as the antilog of the intercept and slope values, respectively, while ΔG was evaluated by using *K_b_* value in van’t Hoff equation.

The values of all binding parameters are provided in Table 3. The ‘*K_b_*’ values were calculated in the same order as done in UV-visible spectroscopy, i.e., IST-02 > IST-04 > IST-03 > IST-01. This confirmed the relatively more stable and stronger interaction of IST-02 with DNA. As the binding site size for all compound–DNA complexes was greater than 1 (*n* > 1), it showed that along with intercalation, other types of reversible interactions such as groove binding or electrostatic interaction may be present due to more site availability. Gibbs free energy changes (Table 3) were also in agreement with the values calculated in UV-visible spectroscopy.

#### 2.3.3. DNA Binding Studies by Cyclic Voltammetry

DNA binding predictions by spectroscopies (UV-visible, Flu-) could further be complemented by cyclic voltammetry that describes compound–DNA interaction in terms of changes in fundamental electrochemical parameters (current/potential). The variations in the cyclic voltammograms of all compounds after DNA titrations, at physiological conditions (pH 7.4, 37 °C), are shown in Figure 10. At first, the individual scanning was performed for IST-01–04 (−1.5 to +1.5V) and it was found that each compound exhibited irreversible cyclic process with a reduction peak within the scan range of −0.5 to −1V. The system irreversibility remained unchanged after DNA titrations for each compound. However, the reduction peak current significantly dropped down to 40.61%, 24.21%, 23.47%, and 20.74%, respectively, along with a positive shift in peak potential. Such voltametric observations showed that all tested compounds interacted with the DNA.

The reduction peak current changes were further implied in the following equation for the calculation of binding constants of all tested compounds [41].
(5)$$I_{p}^{2}=\frac{1}{K_{b}[DNA]}(I_{p_{o}}^{2}-I_{p}^{2})+I_{p_{o}}^{2}-[DNA]$$
where *I_po_* and *I_p_* represents the peak current in the absence and presence of DNA, respectively. The binding constant (*K_b_*) was determined by using the above equation in a graph of *I_p_*^2^ vs. *I_po_*^2^ − *I_p_*^2^/[DNA] (Figure 11). The evaluated values of *K_b_*, along with ∆G, are displayed in Table 3.

Since current variations are related to the concentration of the redox species, by adjusting the current values without (I) and with DNA (I_DNA_) in the following equation, in place of the concentrations ratio of the compound–DNA complex (C_b_) and free compound (C_f_), the binding site sizes (n) were calculated for all compounds [36,37,40,41].
C_b_/C_f_ = I − I_DNA/_I_DNA_ = *K_b_* [DNA]/2n(6)
where base pair concentration of DNA is represented as [DNA]/2n. From the plot of I − I_DNA_/I_DNA_ vs. [DNA], the binding site sizes (n) were evaluated from the slope values after substituting the values of the binding constant *K_b_* (Figure 12). The evaluated values are given in Table 3 and are consistent with those obtained from fluorescence data.

All binding parameters (*K_b_*, ∆G, n) obtained spectroscopically and electrochemically complimented each other and indicated that all compounds have a significant attraction for DNA via mixed mode, with preferably greater binding interactions of compound IST-02 and then IST-04 with the DNA.

To further assure the compound–DNA complex formation, the irreversible redox activity of all compounds were scanned at different scan rates (0.1–0.13 V/s), with and without DNA. The overlay of cyclic voltammograms at various scan rates are provided in Appendix A. The current variations with the square root of scan rates were plotted for all compounds and their DNA-bound complexes. The graphs are shown in Appendix A. The data was further implemented in Randles–Sevcik Equation (7) to evaluate the diffusion coefficient (D_o_) of all compounds and their DNA-bound complexes.
(7)$$i_{p}=2.99\times 10^{5}n{\left({\alpha n}_{a}\right)}^{1/2}{\mathrm{AC}}_{0}^{*}D_{0}^{1/2}\upsilon^{1/2}$$

The calculated values of the diffusion coefficient are presented in Table 4. They clearly indicate slow diffusion, with lower diffusion coefficient values for compounds with DNA due to the formation of bulky compound–DNA complexes [42].

#### 2.3.4. DNA Binding Studies by Viscometry

Another method to investigate compound–DNA interaction is to determine the viscosity of DNA in the presence of a compound’s concentration. Viscosity of DNA increases when the planar portion of a drug/drug-like molecule is inserted into the DNA binding pockets and stacked between the base pairs resulting in the widening of the DNA helical structure. This insertion of the planar part is referred as intercalation and the increase in DNA viscosity arises due to the interaction of electronic density of the intercalating part of a compound with the stacked DNA base pairs. DNA is a natural biopolymer, hence the changes in DNA viscosity will be in minute fractions due to the smaller size of the inserted molecule in comparison to DNA. A mixed binding mode is predicted if a constant behavior is observed after a linear rise in DNA viscosity [36]. On the other hand, if DNA becomes denatured, then a drop in viscosity is observed [43]. Denaturation can occur by a variation in pH, rise in temperature or by using organic solvents such as DMF and DMSO [43].

The viscosity of a DNA solution was first measured at its fixed concentration of 10 µM and then after the addition of 10 to 90 µM of each compound, separately. These changes in DNA viscosity were shown in (η/η_o_)^1/3^ vs. [Compound]/[DNA] plots for all compounds (Figure 13). They indicated that, initially, the relative specific viscosity increased linearly before becoming constant after certain concentrations of the compound. The graphical trend in viscosity verified that all compounds exhibit mixed binding modes for compound–DNA interaction, further supporting the docking investigations for binding interactions via partial intercalation and groove binding.

### 2.4. Cytotoxicity Analysis by Cell Line Studies

The in vitro anticancer potential of the tested compounds IST-01–04 was carried out on three cancerous cell lines, namely HeLa (cervical carcinoma), MCF-7 (breast carcinoma), and HuH-7 (liver carcinoma). None of the compounds showed significant cytotoxic activity against cervical (HeLa) and breast cancer cells (MCF-7), suggesting that these drugs might not be potent against these specific cancers (Appendix A). However, among all compounds, IST-02 and IST-04 have shown significant cytotoxic activity against the liver cancer cell line (HuH-7), shown in Figure 14. IST-02 has shown more than 50% cytotoxicity against HuH-7 at 50 ng/µL to 500 ng/µL concentrations, while IST-04 has shown more than 50 percent cytotoxicity at all tested concentrations, i.e., 10 to 500 ng/µL (Table 5).

The cytotoxic effects of IST-02 and IST-04 were also tested on normal HEK-293 (human embryonic kidney) cell line, with both compounds also showing dose-dependent cytotoxic activity against normal HEK-293 cells (Figure 15). However, the percentage cytotoxicity remained less than 50% at concentration range of 10 to100 ng/µL (*p* < 0.001) for both the compounds on HEK-293 cell line. At the same concentrations, the percentage cytotoxicity on the HuH-7 cancer cell lines was greater than 50%, indicating the potential of these drugs being selectively cytotoxic against hepatocellular carcinoma cells (Table 5).

The half maximal inhibitory concentration (IC_50_) values, as given in Table 5, also indicated that both IST-02 and IST-04 have greater anticancer potency against HuH-7 hepatoma cell line. IC_50_ values indicate that a specific dose of a compound is essential to kill 50% of the cancerous cells. Most of the time, IC_50_ values for a cancerous cell line are less, indicating that a lower concentration of drug is needed to kill the cancerous cells, whereas larger values of IC_50_ on a healthy cell line indicate that a higher dose of drug is needed for the death of healthy cells. This type of phenomenon corresponds to the least side effects of a drug. In the present research, IST-02 and IST-04 had IC_50_ value of 3.07 ± 9.47 and 14.60 ± 2.49 µM, respectively, which suggests that these drugs are sufficiently powerful enough to kill cancerous cells at lower concentrations. For healthy cells, the values of IC_50_ for these drugs were 488.30 ± 2.44 and 197.80 ± 8.75 µM, respectively, which indicates lesser cytotoxicity for healthy cell.

The IC_50_ values of isatin–dehydroepiandrosterone conjugates (4a, 4d, and 4e) were reported as 28.19 ± 3.59, 13.90 ± 3.91, and 22.26 ± 6.21μM, respectively, against HuH-7 [44]. In another recent study on triazole-linked 1,4-dihydropyridine-isatin scaffolds (N1, N2, N13), the IC_50_ values were evaluated as 13.66, 6.73, and 19.36 μM, respectively, for the same cell line [45]. The extract of *Isatis indigotica* leaves and indole alkaloids (Ih, Ij) were tested on the same cell line and the IC_50_ values were reported as 58 μM [46] and 4.88 ± 0.78 and 6.60 ± 1.20 [47], respectively. The IC_50_ values evaluated for isatin derivatives in current studies have been low compared to the literature cited above, especially for the compound IST-02. These results indicate that our compounds are more potent than previously reported derivatives/compounds against liver cancer. To further validate these in vitro results and to authenticate the anticancer potential of tested compounds, in vivo experiments on animal models of respective cancers are required in future.

## 3. Experimental

### 3.1. Materials and Methods

Analytical grade reagents and chemicals were used throughout the experimental procedures. Solvent purification and drying were performed using standard methods. Tetramethyl silane (TMS) was used as an internal reference and each compound and TMS were dissolved, separately, in deuterated solvents to record NMR spectra. For the expression of chemical shift values, the parts per million (ppm) unit was used. For binding studies, salmon sperm DNA was used as received from Sigma-Aldrich. To prepare the stock solution of DNA, deionized water was used. The DNA solution was further diluted to record its absorbance at λ_max_ of 260 nm on a UV-visible spectrophotometer. The Beer–Lambert law was used (C = A_260_/ε × l, where ε is 6600 cm^−1^ M^−1^ and l is 1 cm), and the concentration of DNA was determined to be 5 × 10^−5^ M. The absorbance ratio A_260_/A_280_ was calculated as 1.6, which ensured that the DNA solution was free of other cell organelles including proteins and hence was pure for DNA binding studies [48]. The desired µM solutions of DNA were prepared and their µL quantities were further employed in spectrophotometric and cyclic voltametric DNA binding titration experiments. Contrarily, in viscosity experiments, increasing compound concentrations (µM) were added into a DNA fixed concentration. For cytotoxicity evaluation studies, human cervical (HeLa), breast (MCF-7), and hepatocellular (HuH-7) cell lines were cultured for cancer cell line activities. In addition, human embryonic kidney cell line (HEK-293) was cultured for normal cell line activity in dulbecco modified eagle media (DMEM)–high glucose with 10% FBS (Thermo Fischer Scientific, Waltham, MA, USA) and 1% pen-strep MTT (3-[4,5-dimethylthiazol-2-yl]-2,5-diphenyltetrazolium bromide (Sigma-Aldrich, St. Louis, MO, USA).

### 3.2. Instrumentations

Gallenkamp melting point apparatus was used to determine the melting point via open-end capillary technique. IR analysis was performed on a Bruker FT-IR Bio-Rad-Excalibur spectrometer bearing series mode no. FTS 300 MX. To acquire ^1^H-NMR of the synthesized compounds, a Bruker 300 MHz NMR spectrometer was used. A 75MHz NMR channel was used to obtain ^13^C-NMR decoupled spectra of compounds using proton decoupling frequency. For DNA binding titration experiments, a UV-visible spectrophotometer (Shimadzu-1800, Series mode no. TCC-240 manufactured in Japan), fluorescence spectrophotometer F-7000 model FL2133-007, and AUTOLAB PGSTAT-302 (Metrohm, manufactured in Netherlands) version 4.9 of GPES software were used. The cell temperature of each instrument was controlled at 37 °C with the help of a temperature monitoring device. Wavelength scan, emission scan mode, 2400 nm/s scan speed, 0.0 s time delay, 400 V PMT voltage, 0.5 s response time, 5.0 nm each EX-slit and EM slit, 200.0 nm EM start wavelength, and 900 nm EM end wavelength were the instrumental parameters used in the fluorescence spectrophotometer. For cyclic voltammetric investigations, a conventional double-walled cell was used containing 99.9% pure platinum wire with 0.5 mm diameter as the counter electrode, glassy carbon with diameter 0.3 mm as the working electrode, and a saturated calomel (SCE) filled with 3.5 M potassium chloride (KCl, saturated) as the reference electrode. To obtain a shiny surface of glassy carbon electrode, soft rubbing of the electrode was performed on alumina slurry. It was later washed with distilled water and further sonicated for 30–45 s to remove any type of contamination. A digital viscometer bearing model no. AVS 310 (Schott Gerate automated) was used for viscosity measurements.

### 3.3. Synthesis of Isatin Derivatives

The synthesis protocol for the preparation of 5-chloroisatin-based heterocycles is shown in Figure 1. In the first step, 5-chloroisatin (**1**) was treated with benzyl bromide (**2**) to synthesize the product (**3**) which was further reacted with thiosemicarbazide (**4**) to obtain Schiff base product (**5**). In the last step, Schiff base product was reacted with alpha-halogenated compounds to yield desired compounds 1-benzyl-5-chloro-3-(2-heteroarylhydrazono) indoline-2-ones (IST-01, IST-02, IST-03, IST-04). The first reaction was carried out in anhydrous DMF under an inert atmosphere with a constant reflux of 12 h, while the second step was accomplished in the presence of dry ethanol and under controlled temperature and reflux of 3 h. The problem of side products formation in the final step was minimal as the reaction was carried out in dry distilled ethanol. The reactions were supervised by aluminum TLC plates having silica gel coated with F_254_ indicator using mobile phase ethyl acetate:*n*-hexane (4:6). All synthesized derivatives were analyzed by FT-IR, ^1^H NMR, and ^13^C NMR spectroscopic techniques.

### 3.4. Characterization Data

#### 3.4.1. (Z)-1-Benzyl-5-chloro-3-(2-(4-(2-oxo-2H-chromen-3-yl)thiazol-2-yl)hydrazono)indolin-2-one (IST-01)



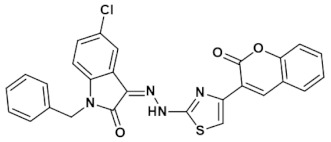



Yield = 89%; Light yellow powder; M.P = above 300^o^C;R_f_ = 0.79 (ethyl acetate: *n* hexane, 4:6);FT-IR (ATR) in cm^−1^, 3233 (H-N), 3119 (H-C, Ar), 1738 (C=O), 1671 (C=O), 1549 (C=N, imine), 1441 (C=C, Ar); ^1^H-NMR (300 MHz, CDCl_3_); in δ (ppm), 13.34 (s, 1H, N-H), 8.63 (s, 1H, coumarin-H), 8.10 (s, 1H, thiazoline-H), 7.67(d, 1H, *J* = 2.1 Hz, isatin-H), 7.64 (s, 1H, isatin-H), 7.59–7.53 (m, 4H, coumaryl-H), 7.41–7.32 (m, 5H, phenyl-H), 7.23 (d, 1H, *J* = 2.1 Hz, isatin-H), 5.01 (s, 2H, methylene-H), ^13^C-NMR (75 MHz, CDCl_3_) in δ (ppm), 165.61 (thiazole, C=N), 161.63(coumarin C=O), 152.99 (isatin C=O), 145.08 (imine, C), 139.77, 139.18, 134.86, 131.50, 129.61, 129.50, 129.04, 128.89, 128.47, 128.13, 127.34, 124.66, 120.42, 119.58, 116.38, 113.85, 110.84, 43.58 (methylene, C); MS (*m/z*): 512.07 (100%).

#### 3.4.2. (Z)-Ethyl 2-(3-((1-benzyl-5-chloro-2-oxoindolin-3-ylidene)amino)-4-oxo-2-thioxoimidazolidin-1-yl)acetate (IST-02)



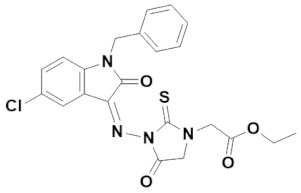



Yield = 45%;Brownish powder; M.P = 200–201 °C;R_f_ = 0.58 (ethyl acetate: *n* hexane, 4:6);FT-IR (ATR) in cm^−1^, 3057 (H-C, Ar), 2905 (C-H SP^3^ Stretching), 1721 (C=O), 1691 (C=O), 1650 (C=N, imine), 1509 (C=C, Ar); ^1^H-NMR (300 MHz, CDCl_3_); in δ (ppm), 8.03(d, 1H, *J* = 2.4Hz, isatin-H), 7.34–7.28 (m, 5H, phenyl-H), 7.26 (s, 1H, isatin-H), 7.22 (d, 1H, *J* = 2.4Hz, isatin-H), 4.97 (s, 2H, thiohydantoin-H), 4.70 (s, 2H, CH_2_), 4.30 (q, 2H, *J* = 7.2Hz, CH_2_), 4.01(s, 2H, CH_2_), 1.31(t, 3H, *J* = 6.9Hz, CH_3_), ^13^C-NMR (75 MHz, CDCl_3_) in δ (ppm), 172.00 (thiohydantoin, C=S), 171.36 (thiohydantoin, C=O), 166.21 (ester C=O), 163.97 (isatin, C=O), 148.08 (imine, C), 143.16, 135.03, 132.23, 128.94, 128.63, 128.10, 127.95, 127.37, 118.00, 110.53, 62.38, 44.29, 43.84, 32.80, 14.18; MS (*m/z*): 453.13 (100%).

#### 3.4.3. (Z)-1-Benzyl-5-chloro-3-(2-(4-(3-oxo-3H-benzo[f]chromen-2-yl) thiazol-2-yl)hydrazine)indolin-2-one (IST-03)



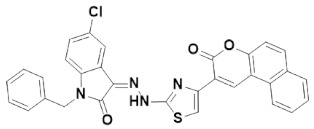



Yield = 86%;Light yellow powder; M.P = above 300 °C; R_f_ = 0.81 (ethyl acetate: *n* hexane, 4:6);FT-IR (ATR) in cm^−1^, 3188 (H-N), 3012 (H-C, Ar), 1691 (C=O), 1508 (C=O), 1455 (C=N, imine), 1434 (C=C, Ar); ^1^H-NMR (300 MHz, CDCl_3_); in δ (ppm), 12.88 (s, 1H, N-H), 7.81 (s, 1H, coumarin-H), 7.67(d, 1H, *J* = 2.1 Hz, isatin-H), 7.58 (s, 1H, thiazoline-H), 7.31 (s, 1H, isatin-H), 7.36–7.29 (m, 6H, coumaryl-H), 7.28–7.25 (m, 5H, phenyl-H), 6.74 (d, 1H, *J* = 2.1 Hz, isatin-H), 4.96 (s, 2H, methylene-H), ^13^C-NMR (75 MHz, CDCl_3_) in δ (ppm), 180.00 (thiazole, C=N), 161.63(coumarin C=O), 160.80 (isatin C=O), 141.33 (imine, C), 139.07, 139.01, 134.86, 134.47 133.31, 131.50, 131.06, 130.88, 130.09, 129.79, 129.50, 129.04, 128.89, 128.47, 128.13, 127.34, 126.43, 124.66, 121.98, 120.82, 120.42, 119.58, 116.38, 113.85, 63.22, 43.71 (methylene, C); MS (*m*/*z*): 562.09 (100%).

#### 3.4.4. (Z)-Ethyl 2-(2-(1-benzyl-5-chloro-2-oxoindolin-3-ylidene)hydrazinyl)thiazole-4-carboxylate (IST-04)



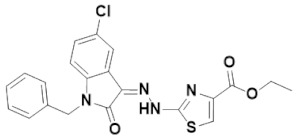



Yield = 80%;Yellow powder; M.P = 192–193 °C;R_f_ = 0.81 (ethyl acetate: *n* hexane, 4:6);FT-IR (ATR) in cm^−1^, 3232 (H-N), 3119 (H-C, Ar), 2970 (C-H SP^3^ Stretching), 1738 (C=O), 1671(C=O), 1637 (C=N, imine), 1549 (C=C, Ar); ^1^H-NMR (300 MHz, CDCl_3_); in δ (ppm), 13.24 (s, 1H, N-H), 7.79 (s, 1H, thiazoline-H), 7.61(d, 1H, *J* = 2.1Hz, isatin-H), 7.38–7.26 (m, 5H, phenyl-H), 7.21 (s, 1H, isatin-H), 7.18 (d, 1H, *J* = 2.1Hz, isatin-H), 4.97 (s, 2H, methylene-H), 4.44 (q, 2H, *J* = 7.2Hz, CH_2_), 1.43(t, 3H, *J* = 6.9 Hz, CH_3_), ^13^C-NMR (75 MHz, CDCl_3_) in δ (ppm), 166.80(ester, C=O), 161.35 (thiazole, C=N), 161.27(isatin C=O), 144.21 (imine, C), 140.10, 134.73, 130.35, 129.83, 129.00, 128.78, 128.08, 127.32, 121.19, 121.06, 120.44, 110.97, 61.55, 43.54, 14.42; MS (*m*/*z*): 440.07 (100%).

### 3.5. Theoretical Procedures for Compound–DNA Binding

#### 3.5.1. DFT (Density Functional Theory)

In this study, all calculations were performed with the Gaussian 09 program using its default criteria [49]. The equilibrium structures were fully optimized without any symmetry constraints by DFT methods at B3LYP level of the theory using 6-311++G(d,p) basis sets. A conformational analysis was performed to obtain the lowest energy structures of the isatin derivatives. The electronic parameters including HOMO, LUMO, and HOMO–LUMO band gaps were determined as a difference of highest occupied molecular orbitals (HOMO) and lowest unoccupied molecular orbitals (LUMO) energies. Output structures of isatin derivatives were visualized using GaussView.

#### 3.5.2. Molecular Docking

Docking studies were performed on the crystal structure of berenil–dodecanucleotide complex at a resolution of 2.5 Å, as obtained from Protein Data Bank (PDB) code 2DBE. Structure energy was minimized by using Amber10 forcefield. The active site of DNA was selected using MOE 2015. The selected binding region contained all the residue of the active site reported in the up-to-date literature. Moreover, the compounds were also prepared by refining the chemical correctness (3D protonation), ionization, and stereo-chemical variation. The energy minimization was achieved at physiological pH of 7.4. Molecular docking of inhibitors within the active site generated maximum 5 poses per ligand where each pose was assigned a score by MOE. The pose having highest score has the most favorable conformation for the ligand. The best pose based on the score was selected.

### 3.6. Experimental Procedures for Compound–DNA Binding

#### 3.6.1. UV-Visible and Fluorescence Spectrometric Experiments

UV-visible and fluorescence spectrophotometric investigations were performed for each compound (IST-01–04), individually, and then in presence of DNA concentrations at physiological pH (7.4) and temperature (37 °C) [36,37,40,41,42]. The individual spectrum of each compound was recorded at optimized concentration (5 × 10^−6^ M), then DNA (10–80 µM) titrations were carried out with each compound, separately. Prior to each spectral run, the establishment of an equilibrium for compound–DNA adduct was assured by keeping the reaction mixture in a cuvette to rest for at least 4–5 min at physiological temperature (37 °C).

#### 3.6.2. Cyclic Voltammetric Experiments

Cyclic voltametric (CV) experiments were run under similar conditions as set in photophysical titration experiments. CV responses of the tested compounds before and after DNA titrations were recorded within the potential range of −1.5 to +1.5 V at 100 mV/s scan rate. For the diffusion coefficient calculation of each compound and its compound–DNA complex, the scanning of the CV responses was performed in the scan range of 30–150 mV/s, with a difference of 20 mV/s. Oxygen was completely flushed out from the system by bubbling argon gas for 5–8 min, while to maintain the cell at 37 °C, a continuous water circulation within the double wall of the cell was provided by a water-circulating bath at this temperature.

#### 3.6.3. Viscosity Experiments

DNA viscosity (η_o_) was measured at a fixed concentration of 10 µM. In contrast to UV-visible, Flu- and CV experimental work, the titration was carried out by using varying concentrations of each compound, separately, in the DNA solution. Changes in DNA viscosity were measured as (η). The relative viscosities were calculated from η_o_ and η values and their cube roots were plotted against compound and DNA concentration ratios.

### 3.7. Cell Line Assays

Cell line studies of tested compounds (IST-01–04) were carried out on three cancerous cell lines, namely cervical cancer (HeLa), breast cancer (MCF-7), and liver cancer (HuH-7), and one normal human embryonic kidney cell line (HEK-293). The percentage cytotoxicity of all compounds for cancerous as well as healthy cell lines were evaluated by dose-dependent MTT analysis [4,5,32]. The Huh-7 and HeLa cells were grown in DMEM (high-glucose medium) supplemented with 10% fetal bovine serum (FBS) and 1% penicillin–streptomycin (pen-strep), whereas the HEK-293 and MCF-7 cells were cultured in RPMI (high-glucose medium) supplemented with 10% fetal bovine serum (FBS) and 1% penicillin–streptomycin (pen-strep). Exponentially growing cells were counted and 10,000 cells per well were plated, in triplicate, in flat-bottomed 96-well plates (Nunc, Roskilde, Denmark). The volume of the cells was kept at 100 µL per well. Each tested compound was dissolved in supplemented media and 10% DMSO, separately, to obtain different concentrations (1.0 ng/µL, 10 µg/µL, 50 ng/µL, 100 ng/µL, and 500 ng/µL). Each concentration of the drugs was added to the 96-well plate, to obtain a final volume of ~200 µL/well. In addition, each concentration was tested in triplicate on all HeLa, MCF-7, HuH-7, and HEK-293 cell lines. Control wells contained solvent control (without drug) and blank media (without cells). Subsequently, 5 mg/mL of MTT was dissolved in 1 mL PBS. Accordingly, 15 µL of prepared MTT solution was added to each well and incubated for 3 h at 37 °C, so that intracellular purple formazan crystals became visible under microscope. Following the formation of formazan crystals, all the solution from each well was removed. Then solubilizing solution, i.e., 150 µL DMSO, was added in each well. The plates were left at room temperature for a few minutes while DMSO solution was mixed thoroughly by pipetting up and down to dissolve the formazan crystals. Finally, the absorbance of the cells was measured by spectrophotometer at 550 nm. From the obtained data of absorbance, % cell viability was calculated using the following equation.
% age of cell viability = (A_sample_ − A_blank_)/(A_control_ − A_blank_) × 100(8)

## 4. Conclusions

Four new isatin derivatives, IST-01–04, have been synthesized and characterized for their structures by various spectroscopic techniques and further examined for DNA binding and cytotoxicity activities for their anticancer potential candidacy. Results obtained from molecular docking and DFT exhibited the receptive nature of IST-02 and IST-04 and their relatively greater binding nature than IST-01 and IST-03. The computed electronic parameters predicted higher charge transfer and hence reactivity for IST-02 and IST-04 due to smaller HOMO–LUMO gap. Theoretical findings were further confirmed by experimental DNA binding investigations by spectroscopies (UV-visible, Flu-) and cyclic voltammetry (CV) under physiological pH of 7.4 and at 37 °C. The trends in spectral and voltametric responses, as well as evaluated binding parameters (*K_b_*, ΔG), indicated that among all compounds, IST-02 and IST-04 have greater and more spontaneous binding affinity for DNA. Binding site size (n) calculations from Flu- and CV data depicted more sites’ availability for interactions and confirmed a mixed type of interactions, as predicted in docking analysis. DNA viscosity measurements in the presence of tested compounds further authorized our findings of mixed binding mode, as assured from theoretical and experimental DNA binding studies. In vitro cell line studies of the compounds revealed that IST-02 and IST-04 have better and selective anticancer potential on hepatocellular carcinoma cells (HuH-7) compared to normal cell lines. Furthermore, these compounds have lower IC_50_ values compared to previously reported derivatives of the same class, indicating their high potency as potential anticancer drug candidates. All findings were in good agreement with each other and thus merit further in vivo testing of these compounds. The overall DNA binding and cytotoxicity results were found to be more significant for the compound IST-02.

## Data Availability

Not applicable.

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
