# Peer review of "Investigations on Anticancer Potentials by DNA Binding and Cytotoxicity Studies for Newly Synthesized and Characterized Imidazolidine and Thiazolidine-Based Isatin Derivatives"

_molecules, 2022, doi:10.3390/molecules27020354_

Round 1

Reviewer 1 Report

Presented manuscript describes sufficiently the anticancer potential of imidazolidine and thiazolidine based isatin derivatives. The work is well organized that some minor errors in text e.g. ± with or without spaces, shall be corrected. Moreover, detailed statistical analysis e.g. standard deviations and statistical significance, shall be emphases both in text and graphs (box and whiskers plot and asterisk). Authors do not explain why the used drug concentrations were chosen. In general the study seems to be a good contribution to pharmacological studies but careful statistics elaboration can improve it and allow for better concluding.

Author Response

We are very thankful to the reviewer for appreciating our work. All necessary revisions have been incorporated within the manuscript and highlighted with yellow highlighter. Some replies are given as follows.

  • ± with or without spaces

Reply: Now all are with space and correction for space, where necessary, is highlighted.

  • Moreover, detailed statistical analysis e.g. standard deviations and statistical significance, shall be emphases both in text and graphs (box and whiskers plot and asterisk).

Reply: Thank you very much for the suggestion. The standard deviations for IC50 values were already included in the Table 5. Statistical significances are now added in text and figure wherever possible and required. Changes are highlighted in yellow.

  • Authors do not explain why the used drug concentrations were chosen…….

Reply: In experimental part “3.6.1 “, it is already given that “Individual spectrum of each compound was recorded at optimized concentration (5 ×10-6 M)”

            In DNA binding spectral and electrochemical studies, first we optimized the compound/drug concentration, so that we could see its binding interactions with varying DNA concentrations in term of changes in spectral (UV-visible, Flu-) and electrochemical (cyclic voltametric) responses. From absorbance, flu- intensity and current/potential changes, DNA binding parameters were obtained using known equations (equation ii to vii) as mentioned in the manuscript at appropriate places.

Reviewer 2 Report

molecules-1513005: Investigations on Anticancer Potentials by DNA Binding and Cytotoxicity Studies for Newly Synthesized and Characterized Imidazolidine and Thiazolidine Based Isatin Derivatives.

The Authors presented an interesting study on four imidazolidine and thiazolidine based isatin derivatives regarding their potential application as an anticancer substance. The study is well planned and all methodologies applied are correct; however, the study lacks tackling several aspects which need to be considered before the paper is ready for publication.

Keywords: Keywords must be improved.

Abstract: HuH-7 liver carcinoma) should be HuH-7 (liver carcinoma) at line 27.

Introduction: There is no need of Figure-1 (isatin structure).

Results and discussion:

Section 2.1. Chemistry of synthesized compounds: Please correct all the typographical errors from line 110-125.

Figure 5 should be improved, as the interactions are not clear.

In table 2, what does it mean "S". Is it binding score? Mention it in the table.

Experimental:

In scheme 1, in the structure IST-01, IST-03 and IST-04, there is no such R2 substituent. There is only R1 substitution. So there is no need to specify each time R2 = H in the scheme.

The stepwise synthetic procedures with every details should must be included in the experimental part, as depicted in the scheme 1.

Mass spectral data should be included for each compound in the section 3.4 Characterization data.

There is no need to write the docking score in experimental data (line 556-557).

Section 3.6. Experimental procedures for compound–DNA binding: Mention the references for each method used to study compound–DNA binding.

Author Response

We are very thankful to the reviewer for appreciating our work. All necessary revisions have been incorporated within the manuscript and highlighted with yellow highlighter. The replies are given as follows.

Keywords: Keywords must be improved.

Reply: Now we tried to improve the keywords and highlighted within the manuscript at appropriate place

Abstract: HuH-7 liver carcinoma) should be HuH-7 (liver carcinoma) at line 27.

Reply: we did it and highlighted within the manuscript at appropriate place

Introduction: There is no need of Figure-1 (isatin structure).

Reply: I will request if you allow for this Figure, as reader of this paper will be with different disciplines of chemistry along with organic chemistry. If you do not agree then I can remove it.

Results and discussion:

Section 2.1. Chemistry of synthesized compounds: Please correct all the typographical errors from line 110-125.

Reply: Necessary typo corrections have been incorporated and highlighted within the manuscript at appropriate places

Figure 5 should be improved, as the interactions are not clear.

Reply: Fig. 5 is now improved, and interactions could be seen clearly.

In table 2, what does it mean "S". Is it binding score? Mention it in the table.

Reply: In table. 2, S refers to scoring function. It has been mentioned now in table. 2 caption and highlighted.

Experimental:

In scheme 1, in the structure IST-01, IST-03 and IST-04, there is no such R2 substituent. There is only R1 substitution. So there is no need to specify each time R2 = H in the scheme.

Reply:  Now R2 is removed from the Scheme 1

The stepwise synthetic procedures with every details should must be included in the experimental part, as depicted in the scheme 1.

Reply: In the experimental section “3.3. Synthesis of isatin derivatives”, some more details of stepwise procedures are incorporated and highlighted.

Mass spectral data should be included for each compound in the section 3.4 Characterization data.

Reply: Mass spectral data is now included for each compound in the section 3.4.

There is no need to write the docking score in experimental data (line 556-557).

Reply: Now the whole sentence is deleted.

Section 3.6. Experimental procedures for compound–DNA binding: Mention the references for each method used to study compound–DNA binding.

Reply: Now references are provided at appropriate place and highlighted as [36,37, 40-42].